# Southern Hemisphere Pressure Relationships during the 20th Century—Implications for Climate Reconstructions and Model Evaluation

**Logan Clark * and Ryan Fogt**

Department of Geography, Ohio University, Athens, OH 45701, USA; fogtr@ohio.edu
* Correspondence: lc491313@ohio.edu

**Abstract:** The relationship between Southern Hemisphere middle and high-latitude regions has made it possible to generate observationally-based Antarctic pressure reconstructions throughout the 20th century, even though routinely collected observations for this continent only began around 1957. While nearly all reconstructions inherently assume stability in these relationships through time and in the absence of direct observations, this stationarity constraint can be fully tested in a model setting. Seasonal pressure reconstructions based on the principal component regression (PCR) method spanning 1905–2013 are done entirely within the framework of the Community Atmospheric version 5 (CAM5) model in this study in order to evaluate this assumption, test the robustness of the PCR procedure for Antarctic pressure reconstructions and to evaluate the CAM5 model. Notably, the CAM5 reconstructions outperformed the observationally-based reconstruction in every season except the austral summer. Other tests indicate that relationships between Antarctic pressure and pressure across the Southern Hemisphere remain stable throughout the 20th century in CAM5. In contrast, 20th century reanalyses all display marked changes in mid-to-high latitude pressure relationships in the early 20th century. Overall, comparisons indicate both the CAM5 model and the pressure reconstructions evaluated here are reliable estimates of Antarctic pressure throughout the 20th century, with the largest differences between the two resulting from differences in the underlying reconstruction predictor networks and not from changes in the model experiments.

**Keywords:** Antarctica; southern hemisphere; pressure; variability; model evaluation; climate change

## 1. Introduction

Although geographically isolated, Antarctica shares strong connections with the climate of the Southern Hemisphere. The Southern Annular Mode (SAM) characterizes relationships between pressure across Antarctica and in the southern midlatitudes [1,2]. In particular, the SAM represents the strength of the pressure gradient across the extratropical Southern Hemisphere (SH) and associated strength of the SH westerly jet [3]. As the dominant mode of SH climate variability [2,4,5], the SAM explains ~20–35% of total Southern Hemisphere monthly atmospheric circulation variability from daily [6] to decadal timescales [7]. Other hemispheric-scale coupled ocean-atmospheric circulation patterns such as the El Niño-Southern Oscillation (ENSO [8]), the Pacific Decadal Oscillation [9]/Interdecadal Pacific Oscillation [10] and the Atlantic Multidecadal Oscillation [11] connect tropical sea surface temperature variations to Antarctica climate via teleconnections associated with tropically-generated Rossby Waves [12–18]. Unsurprisingly, modeling and observational studies note that tropical variability can also influence the state of the SAM [19–24], adding layers of complexity to the connections Antarctica shares with the SH climate.

The relationships between the extrapolar regions of the SH with Antarctica has allowed for numerous observation-based reconstructions of Antarctic climate. In particular, SAM indices have

been generated throughout the 20th century based on mid-latitude pressure observations [25–27]. Indeed, in his SAM index reconstruction, Visbeck [27] assumed that changes in Antarctic pressure were directly proportional to changes in the midlatitudes. As an extension of SAM index reconstructions, the relationship between pressure in the SH mid-latitudes with Antarctica was further employed to reconstruct seasonal mean pressure observations at 18 Antarctic stations back to 1905 [28,29]. Both SAM index reconstructions and the pressure reconstructions at select stations indicate that ozone depletion is an important mechanism for recent changes in the Antarctic atmospheric circulation in austral summer [29–33], consistent with other work [34,35].

While all of these reconstructions add value in understanding historical pressure variability across Antarctica throughout the 20th century, they all are based on a similar underlying assumption. This premise, called the stationarity constraint, is common in nearly all climate reconstructions and assumes that the relationships between the predictor data (in these cases, mid-latitude pressure observations) and the reconstructed climate variable (SAM index or Antarctic pressure observation) remain the same throughout time as they are during the reconstruction calibration period [36]. However, without long-term continuous measurements of the reconstructed variable, the accuracy of this stationarity constraint is nearly impossible to fully assess. As such, the robustness of climate reconstructions in the early 20th century prior to the start of most Antarctic observations [37] is primarily based on relationships/reconstruction skill during the latter half of the 20th century only.

Apart from reconstructions, climate model experiments are important tools for understanding historical Antarctic climate and (for example) have been used to evaluate the effect of the SAM on Antarctic temperature [38], West Antarctic climate variability [39] and Antarctic pressure [32,33]. However, the reliability of climate models in accurately representing Antarctic climate is again challenging to fully determine, as comparison with observations can only be made since 1957 and even shorter for Antarctic sea ice [40]. Given the combination of large interannual variability and the relatively short length of Antarctic climate observations, detection and attribution of Antarctic climate change in both observations and models is difficult [41]. As such, Bracegirdle et al. [42] reiterate the importance of developing longer-term datasets to help quantify the degree to which a climate model may over- or under-estimate responses to various climate forcings in the past, which would provide knowledge of its biases in future projections. This is especially true since other long-term estimates of Antarctic climate variability from gridded reanalyses have been shown to be of lower quality in the early 20th century, directly tied to the quantity of observations assimilated in the reanalyses [43]—there are even notable jumps in the performance of 20th century reanalyses after 1957 for Antarctic pressure [44].

In this study, we provide an alternate form of climate model evaluation as well as a better assessment of the stationary constraint assumed in previous Antarctic climate pressure reconstructions based on statistical relationships with mid-latitude pressure. This evaluation is entirely done within the framework of the Community Atmospheric Model version 5 (CAM5) model. There are multiple benefits of using a non-coupled climate model to perform these evaluations. First, unlike the reanalyses, CAM5 is based only on the prescribed forcing mechanisms and is therefore not sensitive to changes in the quantity of observations [43]. Second, although imperfect, it is a continuous dataset, allowing us to fully examine changes in the strength of the pressure relationship between the mid and high latitudes of the SH and better quantify the overall reconstruction skill purely within the model (since direct data withheld outside a calibration window in the model can be used for comparison). Third, using multiple experiments as in previous work [32,33] allows to determine if any particular prescribed forcing mechanisms significantly influence the mid-to-high latitude SH pressure relationships. It has been suggested that stratospheric ozone depletion has directly influenced the strengthening of the Amundsen Sea Low [45], which is a climatological low pressure center known for its extensive influence on West Antarctic climate variability [46]. Finally, the evaluation can also be thought of as another version of model assessment, in particular when comparing differences in the strength of the relationships between mid and high latitudes pressure over the SH in the model compared to observations, reconstructions and reanalyses.

This paper is structured as follows. The following section describes the data and methods of our study, including more details on the CAM5 model and the procedure for creating the reconstructed pressure datasets. Section 3 analyzes the pressure relationships between the mid and high latitudes of the SH through evaluating the reconstructions conducted entirely within the CAM5 model and describes differences compared to previous work that are related to various forcing mechanisms, model biases or violations of the stationarity assumption. Section 4 summarizes some of the major conclusions and results found in this study, as well as their implications for both climate model assessment and future Antarctic climate reconstructions based on midlatitude data.

## 2. Data and Methods

### 2.1. CAM5

For this study, the National Center for Atmospheric Research (NCAR) Community Atmospheric Model version 5 (CAM5) is configured at a 0.9° latitude × 1.25° longitude horizontal resolution, with a finite volume dynamical core and 26 vertical levels [32]. CAM5 is a non-coupled atmosphere-only climate model, and three experiments representing the main known forcing mechanisms on SH climate variability and change during the entire 20th century are analyzed here to help isolate the sensitivity of various external forcings as in earlier work [32,33]. In addition, each experiment is comprised of 10 ensemble members, each initialized with a random perturbation in air temperature. The first experiment will be termed "Ozone Only," in which ozone concentrations vary over time (1900–2014), whereas SSTs, sea ice concentrations and non-ozone radiative forcings are held to their monthly repeating climatologies. The second experiment is termed "Tropical SSTs + Fixed Radiative," with only time-varying tropical SSTs (1874–2014) prescribed; all other forcings (ozone, greenhouse gases, volcanic activity, etc.) are held to their monthly climatologies in this simulation. The third experiment is termed "Tropical SSTs + Radiative," in which time-varying tropical SSTs and radiative forcings (1880–2014), including ozone, are combined [32]. A list of these experiments can be found in Table 1 below.

**Table 1.** List of CAM5 model experiments, their available time periods and external forcing mechanisms that characterize each experiment.

| CAM5 Experiment | Time Period (Years) | Forcing(s) |
|---|---|---|
| 1. Ozone Only | 1900–2014 | Ozone ($O_3$) <br> Time-varying ozone concentrations |
| 2. Tropical SSTs + Fixed Radiative | 1874–2014 | SSTs (28° N–28° S) <br> Time-varying tropical SSTs |
| 3. Tropical SSTs + Radiative | 1880–2014 | SSTs (28° N–28° S) <br> Time-varying tropical SSTs and all radiative forcings |

### 2.2. Gridded Global Reanalyses

For comparison, mean sea level pressure is also evaluated across the Southern Hemisphere in various 20th century reanalyses. These 20th century reanalyses are unique since the projects were designed to yield an atmospheric circulation dataset based only (or primarily) on the assimilation surface pressure observations, leaving monthly SSTs and sea-ice concentrations as the boundary conditions [47]. The three century-length reanalysis datasets that will be evaluated here include the National Oceanic and Atmospheric Administration/Cooperative Institute for Research in Environmental studies (NOAA-CIRES) 20th Century reanalysis (20CR) version 2c [47], the European Center for Medium Range Forecasting (ECMWF) 20th Century reanalysis (ERA-20C) [48] and ECMWF's Coupled Ocean-Atmosphere Reanalysis of the 20th Century (CERA-20C) [49]. The CERA-20C product is the newest reanalysis that was designed to mitigate some of the flaws observed in ERA-20C [49], with the addition of ocean temperature and salinity from version 4 of the Met Office Hadley Centre (EN4). Each of these three products assimilate surface pressure observations from the International Surface

Pressure Databank (ISPD) [50], but ERA-20C and CERA-20C also assimilate marine surface data (including winds) from the International Comprehensive Ocean-Atmosphere Data Set (ICOADS) [51]. As 20CR and CERA-20C are ensemble reanalyses, we use the ensemble mean MSLP as the best representation of pressure across Antarctica throughout the 20th century.

In addition to the century-length reanalyses, the ECMWF interim reanalysis (ERA-Interim) dataset will be used to compare against our reconstructions. While this dataset only extends back to 1979, previous studies demonstrated its exceptional reliability for Antarctic mean sea-level pressure (MSLP) [52,53]. This allows for another reliable dataset used for the comparison and evaluation of observed differences between the reconstructions and reanalyses.

### 2.3. PCR Reconstruction Procedure and Validation

To evaluate the robustness of Antarctic pressure-based reconstructions throughout the full 20th century, we create reconstructions using the same approach as based on observations, but solely within the CAM5 model. In a similar fashion to previous SAM index and Antarctica pressure reconstructions [26,28], seasonal mean reconstructions of Antarctic station pressure will be conducted using the Principal Component Regression (PCR) technique but instead done entirely within the CAM5 model. To provide the best consistency with Fogt et al. [28], the seasonal mean pressure at 29 midlatitude stations, represented as the closest grid point in CAM5, are used to predict (reconstruct) the seasonal mean pressure at select Antarctic stations (also the closest grid points in CAM5); our reconstruction procedure than uses these model pressure time series to reconstruct Antarctic pressure in the same procedure as Fogt et al. [28]. Prior to performing PCR, the first step in this process involves correlating the midlatitude seasonal mean pressure records to Antarctic pressure records being reconstructed (within the CAM5 model) during the reconstruction calibration period (we use the same period as Fogt et al. [28], 1957–2013). Then, only the midlatitude stations that are significantly correlated at $p < 0.05$ and $p < 0.10$ (5% and 10% networks respectively), are retained. Principal Component Analysis (PCA) is performed on this subset of midlatitude pressure timeseries (known as the predictor dataset) during the calibration period. Along with eigenvectors representing the relationships between the retained predictor data (often called empirical orthogonal functions, EOFs), PCA produces a set of principal components (PCs) showing the time-varying amplitude of the EOFs. Each of these PCs are correlated with the seasonal mean Antarctic pressure record being reconstructed (within CAM5) during the calibration period and again only a subset of these PCs with the strongest correlations are retained for the last step. Finally, these retained PCs are regressed onto the Antarctic pressure record being reconstructed during the calibration period, and the regression coefficients can be used with PCs calculated over the full period (here, 1905–2013) to generate the reconstruction [26,28]. The reconstruction can also be generated by using the relationships between PCs, the regression coefficients and the midlatitude pressure records to generate weights (called beta weights) for each retained midlatitude station [25]. These beta weights can then be employed back in time through the full length of the station observations to generate the reconstruction. In nearly all cases, PCA becomes superior to multiple linear regression by using only a subset of principal components, which acts as a form of noise reduction, whereby only the relationships within midlatitude stations that explain a large portion of the variance in the Antarctic pressure record are used [25,26,28].

The PCR method is repeated for each season and each Antarctic station that was done for the Fogt et al. [28] observation-based reconstructions, as well as every experiment/ensemble member within CAM5. To gauge the sensitivity to the predictor data employed, reconstructions are also generated using the 5% and 10% midlatitude pressure networks separately. Additional reconstructions will be created where trends are already present within the PCR model (original dataset) and where the fitting in the PCR method is based on completely detrended data (detrended dataset) prior to creating the reconstruction. For each Antarctic station, this gives a total of 40 separate reconstructions for each season (and CAM5 experiment), a much larger sample than the 12 reconstructions per station and

season generated in Fogt et al. [28]. A map of the 29 midlatitude stations used as predictor data in the CAM5 model for the reconstruction procedure can be seen in Figure 1 below.

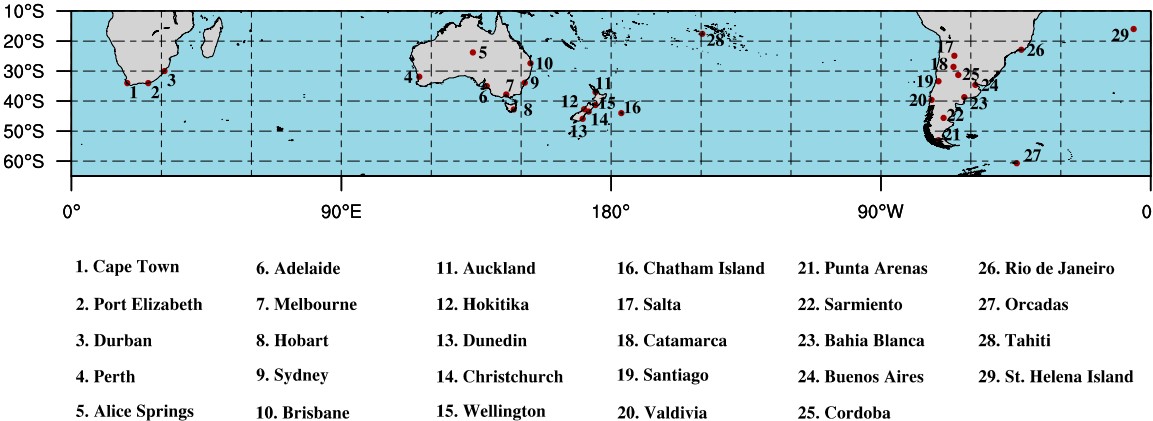

| 1. Cape Town | 6. Adelaide | 11. Auckland | 16. Chatham Island | 21. Punta Arenas | 26. Rio de Janeiro |
|---|---|---|---|---|---|
| 2. Port Elizabeth | 7. Melbourne | 12. Hokitika | 17. Salta | 22. Sarmiento | 27. Orcadas |
| 3. Durban | 8. Hobart | 13. Dunedin | 18. Catamarca | 23. Bahia Blanca | 28. Tahiti |
| 4. Perth | 9. Sydney | 14. Christchurch | 19. Santiago | 24. Buenos Aires | 29. St. Helena Island |
| 5. Alice Springs | 10. Brisbane | 15. Wellington | 20. Valdivia | 25. Cordoba | |

**Figure 1.** Map and list of names of the 29 midlatitude stations used in the study as the grid point locations in CAM5 for performing the PCR reconstruction method.

As in earlier work [26,28], the skill of the model-based seasonal reconstructions is evaluated using metrics such as calibration and validation correlations, reduction of error (RE) and coefficient of efficiency (CE). The calibration correlation is calculated by comparing the reconstructed CAM5 pressure dataset to the original pressure data within CAM5 during the calibration period, as in Fogt et al. [28]. However, the validation period that is responsible for producing our validation correlation is different from previous work. Based on the 'full' reconstructions of Fogt et al. [28], the validation period was also 1957–2013 and a leave-one-out cross validation procedure was employed to generate an independent validation timeseries. Here, since the model data are continuous throughout the entire 20th century, the validation period is the 1905–1956 time period and the reconstructed data during this time can be independently evaluated to the original CAM5 data as a more robust method of evaluating the reconstruction; the validation correlation thus representations the correlation between the reconstruction and the actual CAM5 data during 1905–1956. RE and CE skill metrics are used to further test the reconstruction performance during the calibration and validation periods, respectively [54]. These statistics indicate whether the model-based reconstructions are able to outperform the climatological average for pressure in the calibration and validation periods. RE and CE both range from $-\infty$ to positive 1.0; anything greater than 0.0 indicates that the model reconstructions are providing additional knowledge/skill from simply using the climatological average pressure.

Given the difference in validation periods and approaches afforded to this study by the continuous CAM5 dataset and the fact that the skill metrics of validation correlation and CE are based on comparison of reconstructed with the actual data within the validation period, there will be some differences arising in these metrics compared to Fogt et al. [28]. Nonetheless, the approach here where the calibration period will be used to predict the pressure over the independent validation period serves as an additional means of testing the robustness of the PCR procedure for generating pressure reconstructions, as is typically done for reconstructions with independent validation periods [36,54].

## 3. Results

### 3.1. Reconstruction Performance

As noted before, CAM5 reconstructions were created across the 5% and 10% predictor networks, along with using an original and detrended dataset during the calibration period. Across all of these skill metrics and the 40 reconstructions, the 'best' reconstructions for each station were chosen and the skill metrics were plotted seasonally by experiment as a box plot, along with the best observational

reconstruction values from Fogt et al. [28], in Figure 2. Best reconstructions from the CAM5 data were chosen based on the highest calibration/validation correlations, RE and CE values by station, across all 40 possible reconstructions for each experiment/season. By doing this, every single Antarctic station and season will have a different ensemble member and skill network (5% or 10%) associated with it depending on their performance during the full 1905–2013 time period. This allows us to understand how the best performing CAM5 ensemble members and networks reconstruct pressure through time at different Antarctic station locations, and what the best possible reconstruction skill from CAM5 looks like in comparison to Fogt et al. [28].

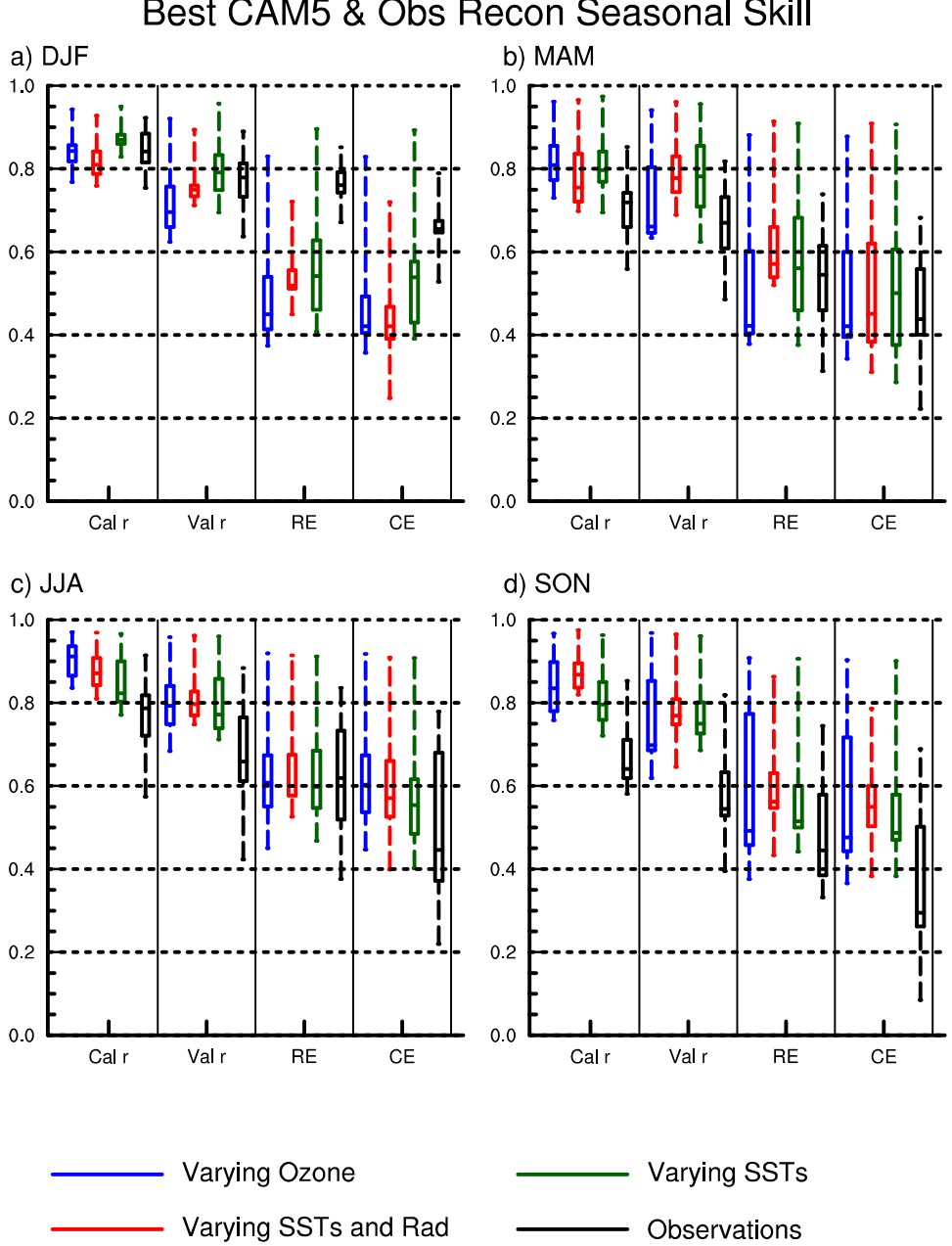

**Figure 2.** Best CAM5 seasonal reconstruction skill metrics across 5%, 10%, original and detrended networks and all ensemble members within each experiment for (**a**) DJF (**b**) MAM (**c**) JJA and (**d**) SON. Best observation-based reconstruction values from [28] are also added in black. See text for additional details on how these skill metrics were calculated for each dataset.

Overall, it was found that the highest skill across all CAM5 experiments is during the austral winter (June–August, JJA), while the observation-based reconstruction from Fogt et al. [28] (black boxplots in Figure 2) performs the best in the austral summer (December–February, DJF), with JJA being the second best season for that dataset. While both the spring (September–November, SON) and autumn (March–May, MAM) seasons exhibit weaker skill between all datasets, the best reconstructions within the CAM5 model tend to outperform the observationally-based reconstructions in every season with the exception of DJF. Nonetheless, all CAM5 experiments, as well as the observational reconstruction, have calibration and validation correlations which generally remain above 0.5, with RE and CE never dropping below zero. Since every skill metric remains positive, the model-based reconstructions created here are thus more reliant than using the climatological average for pressure when examining long-term variability and this preliminary test proves that the PCR method is reliable for estimating 20th century Antarctic pressure variations. This high performance of the CAM5 reconstructions is promising, especially due to the fact that the model has an independent validation period (1905–1957) that does not appear to reduce overall performance with any CAM5 experiment.

To further evaluate the 'best' model-based reconstruction, a timeseries of the original CAM5 data along with the reconstructed seasonal pressure anomalies at select Antarctic stations (within CAM5) is plotted against the observation-based reconstruction in Figure 3. As suggested by the statistics in Figure 2, the correlations of the CAM5 reconstructions with the original CAM5 data (top of each plot) are above 0.50 in all cases; the agreement is especially high at Esperanza (>0.8) on the Antarctic Peninsula, consistent with Fogt et al. [28]. The lower correlations of the CAM5 values with the reconstructions from Fogt et al. [28] do not reflect poor reconstruction skill in either dataset but rather the fact that the model does not contain the full forcing/variability of the real world and therefore does not have the same timing or amplitude of pressure variations, especially those strongly tied to natural variability [32,33]. Notably, CAM5 reconstructions display larger pressure anomalies in MAM and SON, primarily during the first half of the 20th century. Some of the greatest discrepancies between the two reconstructed datasets appears at Esperanza in SON, where both the CAM5 original and reconstructed datasets show pressure anomalies above 11 hPa around 1910, which is in stark contrast to observations which display near-zero and even negative pressure anomalies around that same timeframe. These large anomalies are from the 9th ensemble member and the 5% network at Esperanza in SON, which likely reflect a model bias as the large pressure anomalies in the early 20th century are also seen in the model ensemble mean in SON and especially JJA (cf. Figure 6 of Reference [33]). Overall, the CAM5 datasets (all experiments) typically display larger pressure anomalies (in an absolute sense) for all seasons when compared to the observational reconstruction, which may be driven by the prescribed external forcings within the model framework or the fact that the interannual variability in the observationally-based reconstructions may be slightly underrepresented [29].

## Tropical SSTs + Radiative Pressure Anomalies

**Figure 3.** Pressure anomaly timeseries of CAM5 best reconstruction (same as from Figure 2) along with CAM5 original data for the same ensemble member and the best observational-based reconstruction [28]. Seasonal pressure anomalies are plotted for 3 different Antarctic stations—Amundsen-Scott (South Pole, representing the Antarctic interior), Davis (representing coastal East Antarctica) and Esperanza (representing the Antarctic Peninsula). Correlation values between these datasets are also listed at the top of each figure. Values in black represent the correlation between the best CAM5 reconstruction and the original CAM5 dataset, red values indicate the correlation between the best CAM5 reconstruction and the best observational reconstruction and grey values represent the correlation between the original CAM5 data and the best observational reconstruction. Each value is produced by correlating these datasets over the full time period for each individual station and season.

### 3.2. Evaluating the Stationarity Constraint

The fact that the reconstructions were generated completely independently within CAM5—which is spatially and temporally complete over the entire 20th century—allows us to use the model to more completely analyze reconstruction skill and the underlying assumptions made in generating them. First, to further understand differences between the model-based reconstructions and those from Fogt et al. [28] and as a means of testing the stationarity of pressure relationships between Antarctica and the southern midlatitudes through time, pressure relationships between the model calibration and validation time periods are first examined. Figure 4 shows spatial correlation maps of pressure at select Antarctic stations with pressure in the 15–90°S domain during the validation (left column) and calibration (middle column) period. To assess the stability of these relationships throughout the 20th century, the right column shows the difference (early–late), with statistically significant correlation differences indicated by stippling. These five stations were chosen to provide ample spatial coverage of Antarctica. Figure 4 is based on an ensemble member from the Ozone Only experiment in DJF, as tropospheric ozone forcing is known to have a strong dynamical feedback on Antarctic surface pressure [34,35,55], with ozone-hole induced cooling of the lower stratosphere leading to a delay of the polar vortex [56]. As such, this experiment will likely have the greatest differences between the validation (pre-ozone hole) and calibration (post-ozone hole) period, reflected in SAM index [30,31]

and Antarctic pressure [29,32] variations over the entire 20th century. Since the PCR reconstruction method uses southern midlatitude stations to predict pressure at Antarctic stations, the difference plot in the far right hand column helps to understand the potential uncertainty in the stability of the connection between high- and mid-latitude regions. Reconstructions inherently assume stability in relationships through time, which is why we use CAM5 first here to test this assumption. Importantly, the stationarity constraint is violated when the correlations for the retained midlatitude predictor data locations (which are themselves significant at $p < 0.05$ or $p < 0.10$ during the calibration period) are significantly different in the validation period. It is less concerning for significant correlations to exist in the validation period but not in the calibration period; these locations would not be used in the reconstruction since there was no relationship detected during the calibration period.

As previously noted, connections between Antarctic and the midlatitude atmosphere are driven by large-scale circulation patterns [1,2] and tropical variability [13]. One readily apparent feature in Figure 4 is the large spatial covariance structure of summer pressure across the Antarctic continent, as reflected in the strong positive correlations over most areas poleward of 60° S; the pressure spatial covariance over the continent and the weaker relationship between the continent and the Antarctic Peninsula (easily seen when comparing Esperanza with the other stations), was discussed in previous work [28,32]. In terms of the stationarity constraint, the difference plot in Figure 4 indicates that there are very minute changes through time over the SH continents where the midlatitudes pressure observations used as predictors in our reconstructions reside. The majority of the significant differences (stippling at $p < 0.05$) appear off the west coast of South America and sporadically throughout the SH oceans, with smaller differences observed elsewhere. Overall, Esperanza exhibits weaker correlations to the midlatitudes in DJF, which was found to be consistent in all CAM5 experiments (not shown). Other seasons tend to display smaller differences between the two periods and thus keep this stationarity constraint that is assumed by reconstructed data. Even in DJF where ozone forcing has more extensive influence on Antarctic climate and the relationship between the mid- and high-latitudes weakens slightly in the validation period (left column of Figure 4), southern midlatitude land and Antarctica pressure relationships still remain stable, providing confidence that relationships established during the calibration period are likely stationary prior into the validation period (early 20th century). When examining other experiments, we do note that in the SSTs + Radiative experiment, a few ensemble members suggest there are some significant differences in the pressure correlations in JJA between the Antarctic stations and southern midlatitude regions in South America and parts of southern Africa (not shown). However in the SSTs + Fixed Radiative experiment during JJA, these significant differences over land no longer appear, which indicates that tropical SST forcings may appear to weaken the relationship between the mid- and high-latitudes, primarily during the winter months in at least a few ensemble members but the relationship appears robust in the majority of members and in the ensemble mean, suggested by comparisons in other work [33].

# DJF CAM5 Ozone Forcing-Station Correlation 15-90S

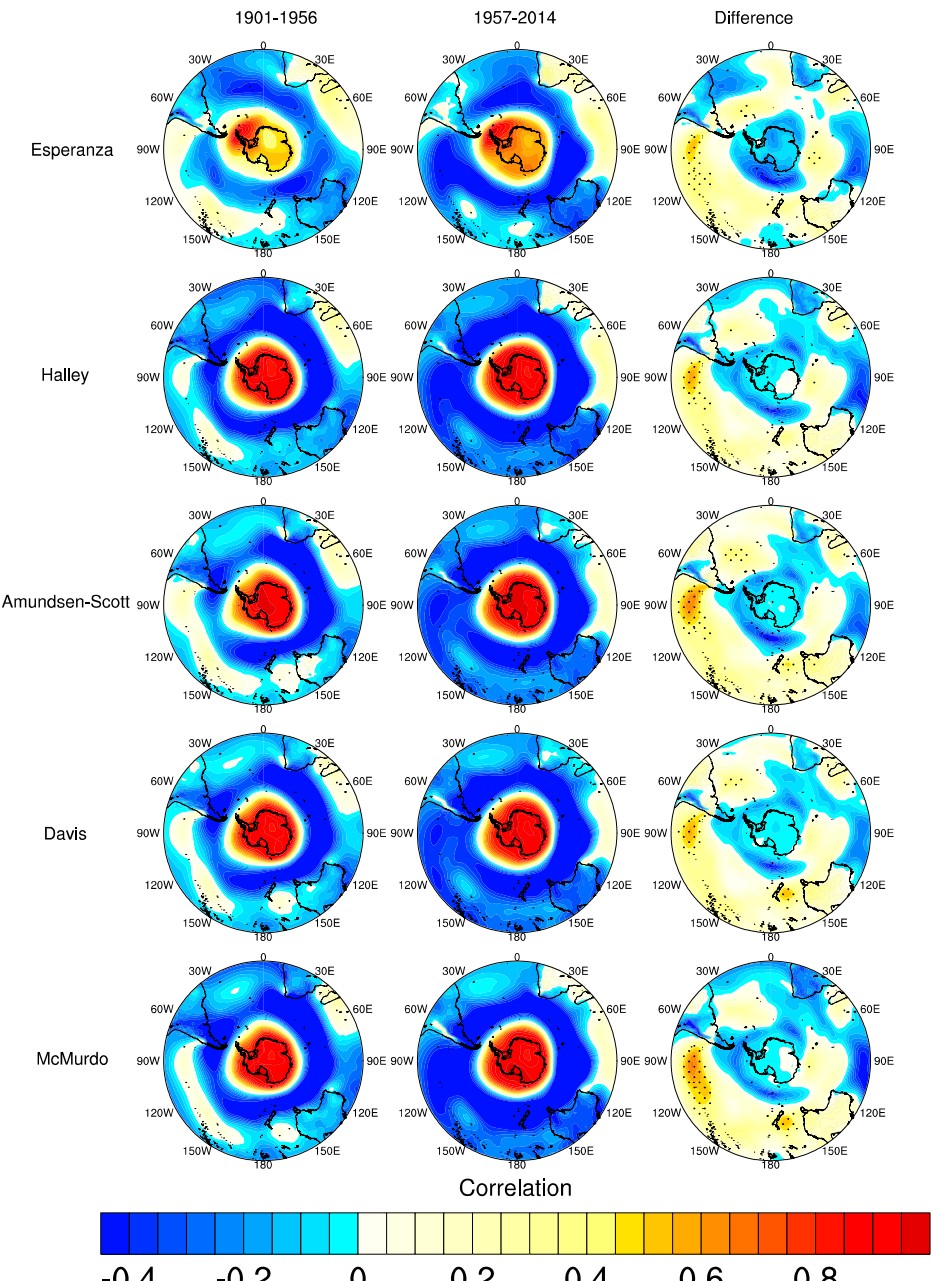

**Figure 4.** Station correlations of 5 locations in Antarctica (Esperanza, Halley, Amundsen-Scott, Davis and McMurdo) to the surrounding high- and mid-latitudes in CAM5. Time periods represent the validation period (1901–1956), calibration period (1957–2014) and a difference of the two (early–late). Stippling on the difference plot indicates regions that are statistically different ($p < 0.05$) from the early period compared to the late period. Ensemble six of the Ozone Only experiment was chosen to represent the overall pattern within CAM5.

Century-length global reanalysis products are an essential tool for predicting/understanding climate variability throughout the 20th century but have only seen notable improvement in the high southern latitudes during the latter half of the 20th century with more routinely collected observations [43]. To understand how some of these newer 20th century reanalyses (and more recently, ERA-Interim) compare to CAM5, which is insensitive to changes in the quantity of observations, we compare the strength of the large-scale atmospheric pressure relationship between the SH

midlatitudes and Antarctica in Figure 5. To assess the sensitivity of various forcings on the SH mid-to-high latitude pressure relationship, the ensemble members from each model experiment are examined separately alongside the reanalyses, plotted in Figure 5 as 30-year running correlations between 35–60° S (midlatitudes) and 60–90° S (high-latitudes); all reanalysis data were detrended over the full 1905–2013 period prior to calculating the correlations. As expected, large discrepancies exist between the CAM5 running correlations and these reanalysis products during the first half of the 20th century. In particular, negative correlation values persist during the full time period in nearly all seasons for CAM5, which is consistent with the strong negative relationships ($p < 0.01$) between Antarctica and the 40–50° S latitudes observed in ERA-Interim and in the SAM [1]. Meanwhile, the century-length reanalyses produce positive and even some strong positive correlation values ($r > 0.70$) during the first half of the 20th century. All CAM5 experiments and ensembles remain fairly stable through time, with only ensemble member 2 in MAM appearing to change sign briefly in the early 1940s, which is still part of the validation period. The strong positive correlation values that depicted by century-length reanalyses during the first half of the 20th century are likely erroneous, as the simultaneous pressure response across the entire SH compromises the reanalyses' ability to conserve mass, a problem noted in earlier work of even contemporary reanalyses [57]. Broadly, Figure 5 demonstrates, consistent with the depiction in Figure 4 for one season and ensemble member, that the pressure relationships between Antarctica and the midlatitudes of the SH are stationary through time, unlike in many other long-term gridded datasets.

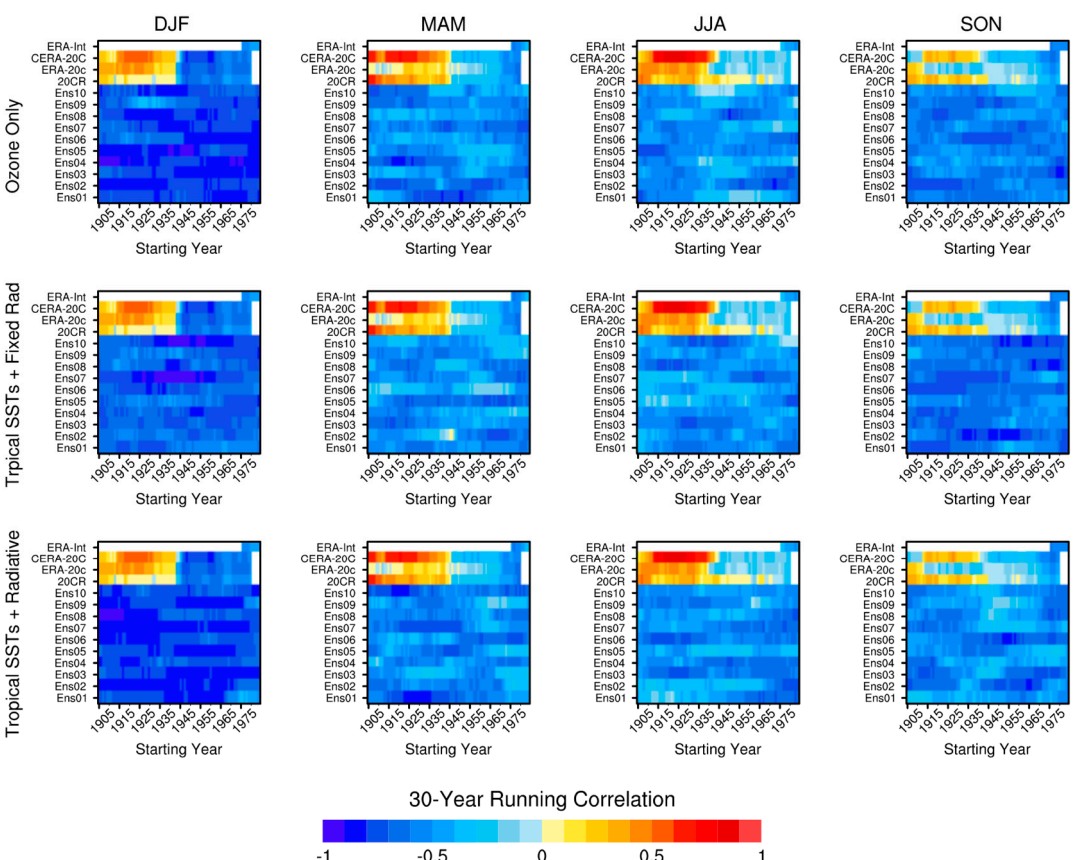

**Figure 5.** 30-year running area-averaged pressure correlations between 35–60° S and 60–90° S for CAM5, ERA-Interim and three century-length reanalyses (CERA-20C, ERA-20C and 20CR). Correlations were done for each CAM5 experiment by ensemble members within each. ERA-Interim does not begin until 1979 and the other century-length reanalyses 30-year trends end in 1980. All reanalysis data were detrended prior to calculating the correlations.

To better understand why these century-length reanalyses produce such large differences in the SH mid-to-high latitude pressure relationships when compared to CAM5, a timeseries of the area-averaged pressure anomalies for JJA and MAM are given in Figure 6 (top two rows). These seasons were chosen for further examination as the century-length reanalyses produced their strongest positive correlations during in the early half of the 20th century in JJA and MAM (Figure 5). The third row on Figure 6 displays the 30-year running correlation for all seasons. Overall, Figure 6 clearly shows how pressure in the reanalysis products for both the mid- and high-latitude regions of the SH has a tendency to behave similarly during the early 20th century, which would explain the strong positive correlation values observed in Figure 5. In particular, periods of positive pressure anomalies in the early 20th century (i.e., 1920s, 1940s) frequently appear in all reanalysis datasets in both the middle and high latitudes of the SH; the Antarctic pressure anomalies in the 1940s reach values up to 10 hPa, consistent with large positive anomalies discussed in earlier work [32,43]. As alluded to previously, this behavior in the reanalyses opposes global mass conservation laws and has been a notable issue in the oceans and high-latitudes prior to 1979, where surface pressure in Antarctica was recorded as being anomalously higher and thus contributed to larger global mean pressure values in products like the 40-year ECMWF reanalysis (ERA-40; [57]). This ultimately provides merit in further examining a globally constrained product (i.e., mass field) to examine teleconnections between the mid- and high-latitudes in order to explain long-term variability where in situ observations are limited [57]. Near the mid-20th century, we begin to see the expected pattern of opposite correlations between the mid- and high-latitude regions in all seasons (bottom row of Figure 6), which would follow expected behavior of pressure covariance structure (i.e., opposite signs) across the extratropical SH, in part reflected in the SAM [1,2,58].

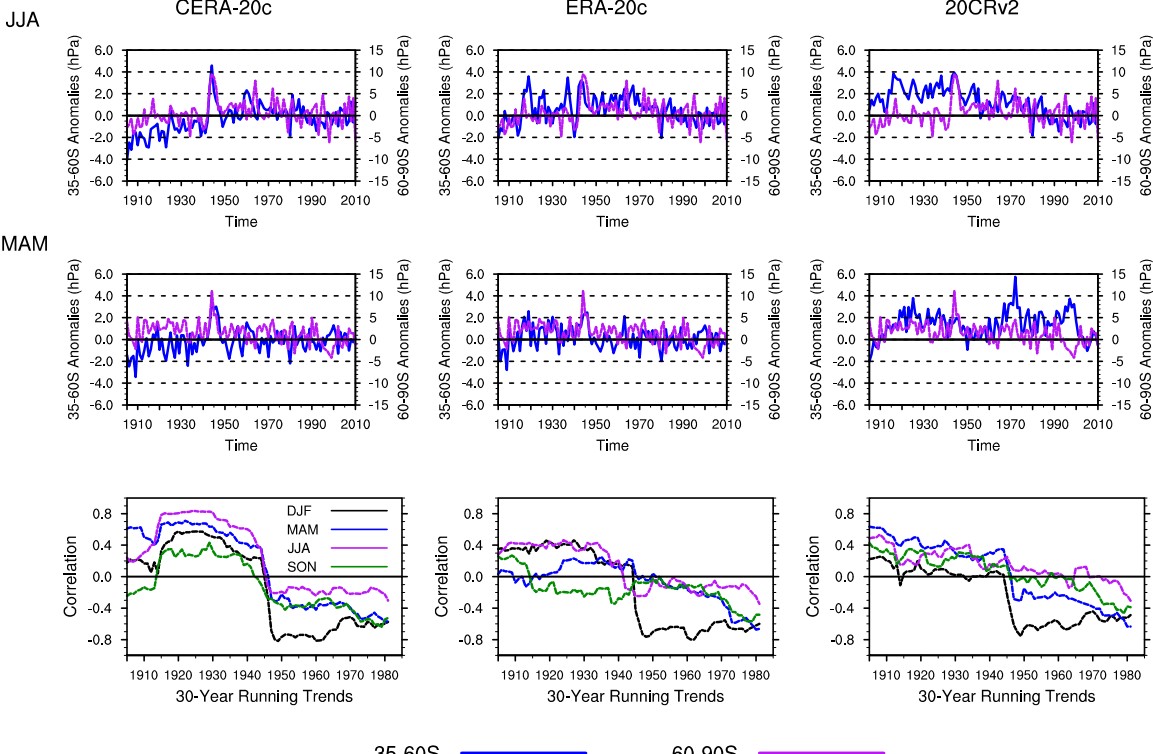

**Figure 6.** Top rows—Seasonal (JJA and MAM) area-averaged (35–60° S, left axis and 60–90° S, right axis) pressure anomalies relative to 1905–2010. Note the different scales on the y-axes for the respective areas. Third row—30-year running pressure correlation between the two regions, by season. Relationships are based on the 20th century reanalysis products—CERA-20C (left column), ERA-20C (middle column) and 20CR (right column).

Although these reanalyses continue to show challenges in their utility during the early part of the 20th century, CAM5 indicates that pressure relationships across the extratropical SH are persevered throughout the early 20th century (Figures 4 and 5). Furthermore, its consistency with observationally-based reconstructions (Figure 2) allow it to be evaluated further to understand potential model biases and the role that external forcings might play on shaping the connection of Antarctic pressure to that in the southern midlatitudes. One additional evaluation of this relationship and the overall Antarctic pressure reconstruction procedure is determining how many midlatitude pressure stations are included in the predictor dataset and if this number itself is stationary outside the calibration period. To analyze this further, Figure 7 displays the number of significantly correlated ($p < 0.10$) midlatitude stations to various Antarctic stations during 30-year moving windows. For comparison, the solid black line on these plots depicts the number of midlatitude stations used in the observationally based reconstructions from Fogt et al. [28].

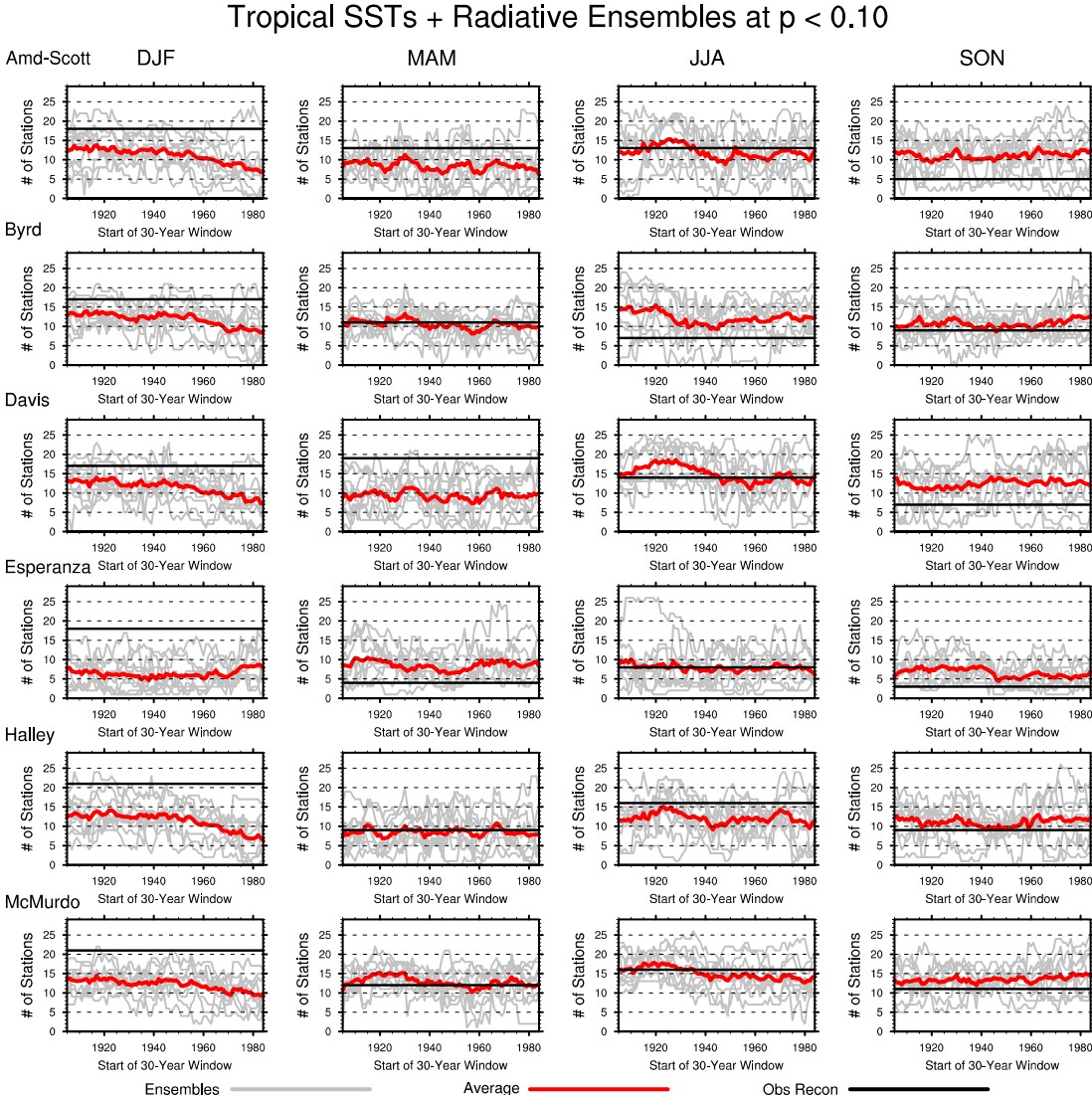

**Figure 7.** Number of midlatitude stations that are significantly correlated at $p < 0.10$ to Antarctic stations in moving 30-year windows. The rows depict the midlatitude pressure relationships for 6 Antarctic stations—Amundsen-Scott, Byrd, Davis, Esperanza, Halley and McMurdo. For each station and season, each ensemble member is plotted (gray lines) from the Tropical SSTs + Radiative CAM5 experiment, with the mean number of stations also shown (red line). The number of stations that were used in the observationally-based reconstruction [28] is indicated by a horizontal black line on each plot.

Interestingly, Figure 7 indicates that in DJF, on average, the 10 ensemble members in the CAM5 reconstructions all have fewer significantly correlated stations through time when evaluated against the observational reconstruction. The reduced predictor network in CAM5 explains the overall weaker CAM5-based reconstruction in DJF compared to the observationally-based reconstruction (Figure 2). It also suggests that at least over the SH continents, the pressure correlations with Antarctica are weaker in CAM5 than in the real-world. In contrast, in SON the model has more significantly correlated midlatitude stations for nearly all 30-year time periods and thus corresponds to the higher skill previously noted within this season (Figure 2). As such, this indicates that reconstruction skill is highly dependent upon this relationship and the size of the predictor set from the midlatitudes, and that a large component of the differences in the model reconstructions and observationally-based reconstructions in Figure 2 stem from differences in the predictor data employed. This is further supported by the fact that the reconstruction skill does not change markedly across model experiments (Figure 2), indicating a weaker role of a forced response in the mid-to-high SH pressure relationship in time, consistent with the similar pattern across experiments seen in Figure 5. Throughout some of the other seasons, natural fluctuations occur with the number of significantly correlated stations but there is no significant change in this relationship over time as suggested by Figure 4. Combining the results in Figure 7 with previous figures, it is apparent the stationarity constraint is not significantly violated, not only broadly in CAM5 (and therefore, likely the real-world), but also at the individual station (regional) scale. We do note that only in DJF does the average number of stations included across the 10 ensemble members (red line) appear to fluctuate in time (left column) in the Tropical SSTs + Radiative experiment. However, as before in Figure 4, the Ozone Only experiment does not show this decrease, suggesting that tropical SST variability may influence the regional extratropical SH pressure relationships through time in DJF. Furthermore, given that the extratropical pressure relationships are already weaker in DJF in CAM5, the changing number of stations cannot fully prove that the stationarity constraint has been significantly violated or that the observationally-based reconstruction reliability is compromised in this season.

Another final assessment of the SH mid-to-high latitude pressure relationships at the regional scale in CAM5 is shown in Figure 8. Here, 30-year running correlations with pressure at the South Pole (Amundsen-Scott) during JJA are displayed for all 29 midlatitude stations that were used as the predictors, plotted separately by ensemble member, as well as the average correlation across all ensemble members. The SSTs + Radiative experiment was chosen to represent CAM5 as it contains the most real-world forcings of all experiments. To facilitate comparison, the midlatitude stations are numbered as in Figure 1 and grouped geographically along the y-axis (separated with solid horizontal black lines and labeled on the right axis). Keep in mind, in order for there to be a violation of the stationarity constraint that would compromise the quality of the reconstruction, the relationships must be significantly weaker in the validation period than they are in the calibration period. At this station-to-station scale, Figure 8 suggests in a few ensemble members, the stationarity constraint may be violated as the relationships established between Antarctica and locations in Australia, South America the Island stations switch sign over the full time period. However, the majority of the ensemble members do not show such dramatic changes in the strength of the station-to-station pressure relationship through time and many ensemble members and regions remain significantly correlated through both the calibration and validation periods or weakly correlated (and therefore not retained) in either the calibration or both periods. Nonetheless, even though some ensemble members appear to violate this stationarity constraint, Figure 2 still indicates that the overall skill in these CAM5 reconstructions remains fairly high, especially in JJA; spurious fluctuations in the relationships on shorter time periods are to be expected due to natural variability. Because the reconstruction statistical model (PCR) is ultimately not based on the individual stations but instead the relationships between them (through the use of PCA on the retained stations), Figure 8 provides further support that the overall stationarity constraint is in fact not violated. For example, at each region and ensemble member, the correlations within and (by extension) across regions remains robust, strongly suggesting that the

retained PCs/EOFs remain robust throughout time regardless of the exact stations retained to represent these patterns.

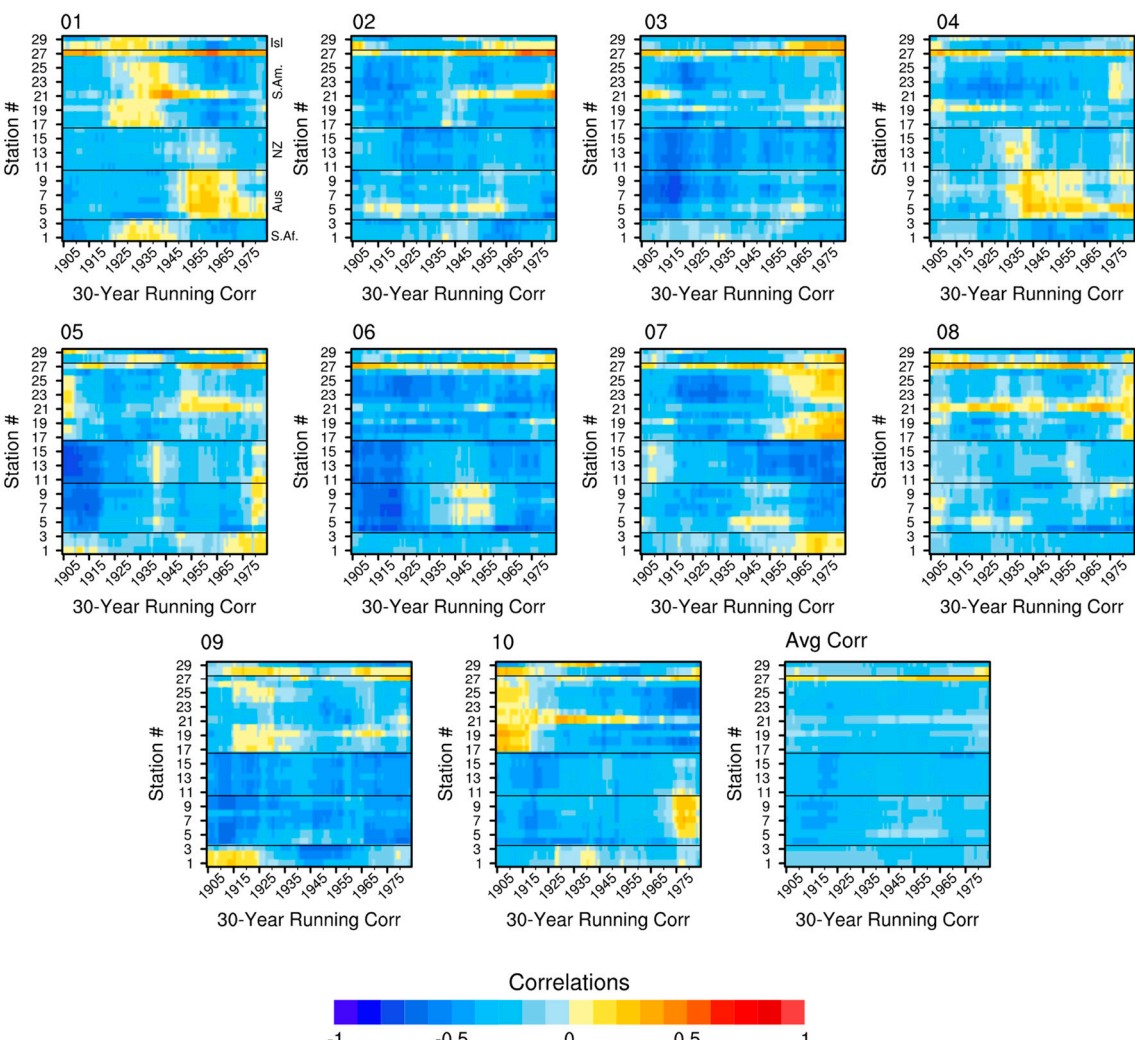

**Figure 8.** 30-year running correlations across all 29 midlatitude stations to Amundsen-Scott during the austral winter for the SSTs + Radiative CAM5 experiment. Each plot depicts an individual ensemble, along with an average between these ensembles in the bottom right. The stations are grouped by region (labeled on right of first plot), with each number on the y-axis corresponding to the station name listed in Figure 1. The southern midlatitude regions include islands (Isl), South America (S. Am.), New Zealand (NZ), Australia (Aus) and South Africa (S. Af.).

The average correlation plot (Figure 8, bottom right) further helps to provide an overall picture of the pressure relationships at fine scales in CAM5. From this figure, on average, the correlations generally remain the same sign (negative) through time, depicting the stationarity of these relationships in the 20th century once again. While this average arises partly from offsetting positive and negative correlations in some regions across ensemble members, the different correlations reflect the changes in natural variability in each ensemble member. It is difficult to assess whether or not this natural variability reflects real-world variability or not, given the problems in other 20th century datasets in the high southern latitudes (Figure 6) [43] and the large natural variability present in observations [41].

## 4. Discussion

The primary goal of creating reconstructions is to build ensembles of new datasets that can estimate historical climate variability in the absence of direct observations. While of significant value at the global scale [36,59], reconstructions are of noted value in regions like Antarctica where meteorological observations remain sparse [60]. However, the short nature of the observational records make it challenging to fully assess the reliability of Antarctic climate reconstructions, especially given the large natural variability present in the high latitudes of the SH [41,61]. The goal of using CAM5 to generate reconstructions in the present study was therefore to not only verify the stationarity constraint that is inherently assumed in reconstructed datasets but to also provide a new perspective on how a model, with the ability to isolate external forcings, behaves in recreating long-term pressure. When evaluated to an observational Antarctic pressure reconstruction [28], our model-generated reconstructions recorded higher skill metrics in all seasons with the exception of DJF. The lower performance in DJF and to some extent the higher performance in SON, appears to be tied to the size of the predictor network (which itself may be due to the spatial resolution of CAM5) and not to various prescribed forcing mechanisms—across all experiments, the model reconstruction performance was similar and in general it performed better than the observationally-based reconstruction when the predictor network was larger (and lower when the predictor network was smaller, as in DJF). Some of the other differences may likely be due to model biases in CAM5, which displays increased variability than the observationally-based reconstruction in JJA and SON, including strong positive pressure anomalies in the early twentieth century. However, it is also possible that the observationally-based reconstruction is underestimating internal variability in the early 20th century. Somewhat surprisingly, the stationarity of the mid-to-high latitude SH pressure relationships were not strongly sensitive to the various external forcings (ozone depletion, tropical SSTs or combined radiative forcings), although these are known to affect large scale modes of climate variability including pressure trends in both the mid and high latitudes [32–34,41]. We interpret this result as these forcing mechanisms influencing both the Southern Hemisphere mid and high latitudes in similar (but often opposite) ways, therefore preserving the statistical relationship between the regions with time, even if the forcing mechanism may change the pressure variability or induce pressure trends in both regions [33]. In general, the model performance and the pressure relationships across the SH were similar in all experiments, with perhaps the SST + Radiative experiment showing the most sensitivity, likely due to tropical SST variability over radiative forcing as the Ozone Only experiment showed little changes throughout the 20th century. Furthermore, in many tests the model results indicated that pressure relationships between Antarctica and the rest of the extratropical SH during the period of Antarctic observations (1957–present) are similar to those observed in the early 20th century (1905–1956). This implies that Antarctic pressure reconstructions based on a stationarity assumption are likely to be robust through the entire 20th century. This is in stark contrast to global 20th century reanalysis products, however, which show marked changes in skill throughout the 20th century [43], including the strong likelihood that these products are not appropriately conserving atmospheric mass in the early 20th century.

The results here are important for climate reconstructions and model evaluations alike, as they independently test reconstruction methods, underlying assumptions and statistical relationships necessarily made in generating reconstructions and the model's ability to simulate these reconstructions. Such work is not new, as climate models have been frequently used to evaluate reconstructions, often by generating 'pseudoproxies' in the degrading model data to represent information contained within proxy records like tree rings or ice cores [36]. Since our work is based solely on measurements of pressure (from observations or the model itself) and not proxy records, the comparisons here are only effectively limited by the skill of the model in representing climate variations and biases within the model and not how to accurately make model data represent proxy data. While the work here suggests differences in the CAM5 pressure compared to the observationally-based reconstruction, the comparisons also suggest overall that the model is an effective tool for understanding 20th century Antarctic climate variability and that published reconstructions [28,29,32,33] of Antarctic pressure

throughout the 20th century are consistent with the range of climate variability depicted within CAM5. Future work is therefore encouraged using these datasets to better understand ongoing Antarctic climate change within a longer-term historical context.

**Author Contributions:** All authors contributed to the analysis, writing and editing of this paper.

**Funding:** This research was funded by National Science Foundation, grant numbers PLR-1341621 and PLR-1744998.

**Acknowledgments:** The Climate Variability and Change Working Group of the Community Earth System Model provided the CAM5 experiments with time-varying tropical SSTs and time-varying radiative forcings. All CAM5 model data can be downloaded from: http://www.cesm.ucar.edu/experiments/cesm1.1/LE. The CERA-20C, ERA-20C and ERA-Interim datasets were gathered from the ECMWF data center online (https://www.ecmwf.int/en/research/climate-reanalysis/browse-reanalysis-datasets), while 20CR version 2c was obtained from NOAA's Earth System Research Laboratory, also available online at https://www.esrl.noaa.gov/psd/data/gridded/data.20thC_ReanV2c.html. Finally, the station-based pressure reconstructions used for comparison in this study are available from https://doi.org/10.6084/m9.figshare.5325541.v3.

**Conflicts of Interest:** The authors declare no conflict of interest.

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
