# Peer review of "Southern Hemisphere Pressure Relationships during the 20th Century—Implications for Climate Reconstructions and Model Evaluation"

_geosciences, doi:10.3390/geosciences9100413_

Round 1
Reviewer 1 Report
Main goal of this study is to examine the stationarity assumption in historic statistical data reconstructions using various model experiments. It is an interesting subject and it can be a good contribution to community dealing with many historical long records.
I would like to suggest this work for publication subject to a revision to address following concerns.
Source of stationarity: It is not clear how these model experiments are selected for this study. What’s the importance of Ozeon, tropical SST, and radiative forcings on the relations between the midlatitude and polar region surface pressure. I expect to see some discussions on the linkage between different forcing and its impacts on the stationarity of the relations. Model experiment: How did radiative forcings are fixed? Both SW and LW are fixed? Only at surface? How about atmosphere? Local SST effects are not considered in the model experiments. What is the role of midlatitude and/or polar ocean temperature? Is observed correlation between midlatitute and polar SLP from the tropical forcings? (See also Comment #8) Lines 152-153, ”significantly correlated stations”: Are correlation patterns comparable to those in observations? Line 248, “actual data”: Is this model data? Line 271, Figure 3 caption, “original data”: Is this ensemble mean? Or, a single ensemble? Figure 3: Correlation between model and observation is very low. At some stations (e.g. Esperanza) show model and observation have opposite trends. I do not see much value to introduce observation reconstruction here. Line 281, “to understand differences between the model-based reconstructions and those from observations”: Same as comment #7, I do not see the value of this comparison. They only share method, but from completely different data set. Also, the model relations between midlatitude SLP and polar SLP would be quite different from those of observations (See also Comment #4). Figure 4. Shouldn’t the center of correlation, particularly for 1957-2014, be collocated with stations used as reference? Except for Esperanza, all other stations correlation patterns are centered at the pole. Also, anomaly maps are all similar. Does that mean all four-station data are highly correlated with others? Lines 342-343, “all reanalysis data were de-trended”: Was de-trended done for the whole period, or for each 30-year window? Lines 420-421, 459-460: Is this because long-term trends from ozone or tropical SST are removed (detrending)? In reality, long-term trend cannot be removed. Conclusion could be different for observation-based reconstructions? Model grid values are spatially smoothed as compared to station observation. Also, model experiments are conducted with smaller sets of forcing. Yet, the reconstruction is not as good as observation-based reconstruction. What could be a reason? Does missing forcing (e.g. midlatitude and polar ocean temperature) play an important role in SLP correlations between tropics and Antarctica?
Reviewer 2 Report
Here, the authors use CAM5 and a network of weather stations to investigate southern hemisphere atmospheric pressure relationships during the 20thcentury. Overall, the manuscript is well-written, scientifically sound and appears to have already gone through careful editing. The authors have done a good job of acknowledging the assumptions and limitations inherent in their study. The figures are clearly presented. In my opinion, this manuscript is ready for publication with only a few minor changes, which are listed below.
Table 1: How did you choose the start times for each model (1874 vs. 1880 vs. 1900)?
Line 111: I assume ozone forcing is included in the tropical SSTs + Radiative experiment, but it would be worth stating explicitly for clarity.
Figure 4: Why not show the other experiments? You only show the ‘ozone only’, but you mention that the SSTs + Radiative experiment may show some indication of non-stationarity, which you dismiss. If space permits, I think it would be helpful to see this output, especially since the pattern in Figure 4 of significant changes in the relationship between high and mid-latitude pressure suggests some changes in the mid-latitude Pacific, which is interesting.
Line 375: Should be ‘alluded”
Line 500: Here you state that SH pressure relationship don’t appear to be influence by any of the forcing mechanisms you assessed. But isn’t there are large body of literature suggesting otherwise? Many of the papers you listed in the introduction have argued that a combination of radiative forcing, ozone depletion and tropical SST changes have lead to changes in the relationship between mid and high latitude pressure in the SH. How do you reconcile these previous findings with your new results that show that none of these factors make much difference?
